# Grape Pomace: A Potential Ingredient for the Human Diet

**DOI:** 10.3390/foods9121772

**Published:** 2020-11-29

**Authors:** Paula Pereira, Carla Palma, Cíntia Ferreira-Pêgo, Olga Amaral, Anabela Amaral, Patrícia Rijo, João Gregório, Lídia Palma, Marisa Nicolai

**Affiliations:** 1CBIOS—Universidade Lusófona’s Research Center for Biosciences & Health Technologies, Campo Grande 376, 1749-024 Lisboa, Portugal; p1204@ulusofona.pt (P.P.); cintia.pego@ulusofona.pt (C.F.-P.); patricia.rijo@ulusofona.pt (P.R.); joao.gregorio@ulusofona.pt (J.G.); lidia.palma@ulusofona.pt (L.P.); 2CERENA—Instituto Superior Técnico (IST Center for Natural Resources and Environment), Universidade de Lisboa, Av. Rovisco Pais, 1049-001 Lisboa, Portugal; 3Instituto Hidrográfico, R. Trinas 49, 1249-093 Lisboa, Portugal; carla.palma@hidrografico.pt; 4Departamento de Tecnologias e Ciências Aplicadas, Instituto Politécnico de Beja, Campus do IPBeja, Apartado 6155, 7800-295 Beja, Portugal; olga.amaral@ipbeja.pt (O.A.); anabela.amaral@ipbeja.pt (A.A.); 5Instituto de Investigação do Medicamento (iMed.ULisboa), Faculdade de Farmácia, Universidade de Lisboa, 1649-003 Lisboa, Portugal

**Keywords:** grape pomace, *Vitis vinifera* L., micronutrients, trace metals, moisture, microorganisms

## Abstract

The industrial production of wine generates annually tons of waste that can and must be properly reused to reduce its polluting load ad increase the availability of passive ingredients to be used in human nutrition. Grape pomace, a by-product of winemaking, beyond being of nutritional value is a bioactive source with high potential value and benefits for human health. Having as main goal the preliminary perception of the potential use of this by-product, the aim of this study was the characterization of eight different grape pomaces. In this sense, ash content, relative ash, moisture, pH, microorganisms, metals (Al, Cd, Cr, Cu, Fe, Hg, Li, Mn, Ni, Pb, and Zn), and semi-metal (As) were reported. The parameter that limits the daily amount ingested of this product is its arsenic content, a non-essential element that belongs to the group of semi-metal. Considering the obtained results and in the light of the restrictions imposed through the legislation in regulations set by the European Commission, the inclusion of grape pomace in the industrial production of foodstuffs could be a step towards the future of human nutrition and health.

## 1. Introduction

Grape pomace is the main by-product of winemaking which may provoke a wide range of changes in climate, environment, and resource capacity [1,2], due to its capacity to polluting the soils and reducing the availability of other ingredients. In the last years, public demand produced modifications in dietary behaviors [3] and ecological lifestyle [4], causing the need for more environmentally safe diets [5,6,7,8]. For all of these reasons, it’s important to keep a balance between human health, mainly the development of potential new ingredients, and environmental impacts [9,10,11].

In 2010, Food and Agriculture Organization (FAO) and Bioversity International organized an International Scientific Symposium about “Biodiversity and Sustainable Diets”, in which an agreement was stretched on a definition of sustainable diets as “those diets with low environmental impacts which contribute to food and nutrition security and healthy life for present and future generations. Sustainable diets are protective and respectful of biodiversity and ecosystems, culturally acceptable, accessible, economically fair and affordable; nutritionally adequate, safe, and healthy; while optimizing natural and human resources” [3,12]. In 2019, the same institution updated this definition of sustainable diets as having four dimensions: (1) nutrition and health, (2) economic, (3) social and cultural, and (4) environmental [12]. In this context, it’s very important to minimize the planet’s environmental impact following sustainable diets and more sustainable food-production practices.

One way to have an eco-friendly food production is recycling the final industrial waste derived from them. Industrial waste, according to its traditional definition, is represented by “unwanted and residual materials generated by industrial activities (manufacturing, mining, control and treatment of process emissions, etc.)” [13]. The poor management and elimination of industrial leftovers have been recognized as a cause underlying serious environmental problem, equally in terms of increasing air contamination, and decreasing water and soil quality [13]. The re-utilization of agro-industrial wastes is a simple and effective manner for fighting against the overthrow in the soil organic carbon, which is the main responsibility of keeping soil quality, with beneficial effects such as decreasing soil erosion and pollutants from natural waters, bettering its quality, rising biomass and productivity, and reducing atmospheric carbon dioxide [14].

Wineries are one of the greatest’s agro-industrial activities worldwide, being also one of the main sectors responsible for producing industrial by-products and waste [15]. According to the 2019 statistical report on the International Organization of Vine and Wine, the wine produced worldwide in the 2018 campaign [16] was 292 million hectoliters, being the highest amount since 2000. Portugal was the 11th worldwide producer, with 6.1 million hectoliters of wine during the 2018 campaign [17].

Due to the size of this sector in the worldwide industry, one of the principal concern is that the winemaking method produces, from top to bottom, abundant diversity of leavings like, vine shoots, grape pomace, wine lees, spent filter cakes, vinasses, and winery wastewater which is important to take care to avoid negative ecological effects [18,19]. Through winemaking, around 25% of the grape mass outcomes in grape pomace which comprises stalks, skin, disrupted cells from the grape pulp, and seeds that remain after the grape crushing and pressing steps get grape juice [19,20,21]. It is projected that the fabrication of 6 L of wine results in about 1 kg of grape pomace, which worldwide consists of 10.5–13.1 million tons annually [22]. Consequently, special attention should also be given to more advantageous options for economics and environment, by the scientific community and manufacturers, with the objective of maximization of the use of all raw resources and leavings resultant from the wine industry trying to reduce to a least its disposal [18]

These surpluses are recognized as a source of various nutritional components, essential and non-essential [22,23] that can be used in the food industry as a source of macro and micronutrients for the manufacture of various new and functional foods for human consumption, creating a vast opportunity for the reduction of waste and a potential source of extraordinary income [24].

World Health Organization (WHO) has reported in 2020 that foodborne diseases can be of infectious or toxic origin, for example from bacteria or environmental pollutants such as heavy metals, this study aims to estimate the health risk of consuming grape pomace flour [25]. Thus, microbiological analysis and the elemental composition were performed. Total bacterial, mold, yeast, fungi, and Enterobacteriaceae were counted. In addition to the percentages in moisture and ash and the pH determination of, the levels of metals concentration such as of aluminum (Al), cadmium (Cd), copper (Cu), chromium (Cr), iron (Fe), arsenic (As), lithium (Li), zinc (Zn), mercury (Hg), manganese (Mn), nickel (Ni), lead (Pb) in different grape pomace variety were also performed to understand whether these flours can be used in the human diet.

## 2. Materials and Methods

### 2.1. Grape Pomace Samples

The grape pomace samples were collected from four different farms, one in Ribatejo (zone 1) and three in Alentejo (zones 2, 3, and 4) in 2017. The samples were identified according to the type of varieties studied: Arinto (A2, A4), Aragonês (Ar2), Merlot (M2), Talia (T1), Syrah (S2), Cabernet Sauvignon (Cs3), Trincadeira (Tr2). Samples were dried in the oven for 24 h at 60 °C, milled in a domestic blade grinder (Moulinex, Alencon, France), and stored in properly sealed propylene bags until analysis.

### 2.2. Microbiological Analysis

For the microbiological analysis, 10 g samples were removed aseptically from each package and were transferred into a sterile plastic package to which 90 mL Ringer’s sterile solution was added. The sample and the Ringer solution were blended for 60 s by using a stomacher (Stomacher^®^400 IUL, Barcelona, Spain) The standard methods, ISO 4833:2003 [26], ISO 21527:2008 [27], and ISO 21528-2:2004 [28] were followed for counting total aerobic plate count (TPC), yeast, and molds count, and Enterobacteriaceae count, respectively.

### 2.3. Moisture Content

Five grams of sample were weighed, transferred into a Petri dish, and then placed in the oven (J.P. Selecta, Barcelona, Spain) with forced air circulation at 105 °C with repeated 4 h cycles until constant weight. All assays were performed in triplicate. The moisture percentage (%) content was determined by the method described by Nicolai et al. [29].

### 2.4. Ash Content

Two grams of sample were placed in the oven (Selecta, Barcelona, Spain) with forced air circulation at 105 °C for 2 h, and after this period it was moved in the muffle furnace (J.P. Selecta-Horn, Barcelona, Spain) for 4 h at 550 °C. The ash content was determined by the method described by Nicolai et al. [29]. All assays were performed in triplicate.

### 2.5. pH Determination

One hundred milliliters of distilled water were added to ten grams of sample. The suspension was agitated, and the pH values were measured, after stabilization of the suspended particles, using a potentiometer (Hanna Instruments pH 211 Microprocessor, Padova, Italy). All assays were performed in triplicate.

### 2.6. Elemental Composition

The determinations of total metal concentrations in samples were made under a quality control regime routinely used in laboratories (recovery of standards and replicates in 3% of the samples). The content of metals was obtained after digestion using a microwave oven (ETHOS PLUS Milestone, Sorisole, Italy) through a three steps procedure (5 min at 100 °C and 250 watts; 10 min at 180 °C and 800 watts and 20 min at 180 °C and 800 watts), using 10 mL of nitric acid. After digestion the concentrations of heavy metals were analyzed by atomic absorption spectrometry (Solaar-Thermo Elemental Thermo Fisher Scientific, Inc., Cambridge, United Kingdom), by flame for Al, Cd, Cr, Cu, Fe, Li, Mn, Ni, Pb, Zn, using a calibration curve obtained from external standards. Arsenic levels were analyzed in a hydride generator, with sodium borohydride added to convert the As(III) to the volatile hydride that was then purged from the solution by a stream of argon gas; again a calibration curve obtained by external standards was employed (Solaar-Thermo Elemental-VP90 Continuous Flow vapor Accessory). Mercury concentrations were directly measured in samples by atomic absorption spectrometry with thermal decomposition, using a Direct Mercury Analyzer (DMA Milestone, Sorisole, Italy). Blanks were prepared using the same procedure without sample. All reagents were of Merck Suprapure quality and pure water was MilliQ grade. Working solutions of metals were prepared using 1.0 × 10^−3^ mg/L (Merck) standard solutions.

All reported uncertainties represent expanded uncertainties expressed at approximately the 95% confidence level using a coverage factor of k = 2. For each analyzed element the limits of quantification and detection [30] as respectively values (in g/kg, in brackets and separated by a comma) as follows: Cu (5.0, 1.7), Cr (5.0, 1.7), Cd (0.5, 0.2), Mn (5.0, 1.7), Fe (5.0, 1.7), Li (5.0, 1.7), Zn (2.0, 0.7), Al (50.0, 16.7), Ni (7.5, 2.5), Pb (10.0, 3.3) and As (5.0 × 10^−2^, 2.0 × 10^−2^) Hg (8.0 × 10^−3^, 3.0 × 10^−3^).

### 2.7. Statistical Analysis

After the sample analysis, statistical analysis was performed seeking to find patterns of metal occurrence within the different types of grape pomace. Varieties were categorized into red and white varieties. Descriptive statistics were performed, and the Mann-Whitney test was used to assess differences in metal concentrations between red (Aragonês, Merlot, Syrah, Cabernet Sauvignon, Trincadeira, and white (Arinto, Talia) varieties. Spearman correlation was used to identify correlations between metal concentrations. The significance level was set at 5%.

## 3. Results

All samples (Table 1) have a moisture content below 10% which means that, due to low humidity, they are naturally preserved from microbial degradation [31,32]. Some of the samples, namely Arinto A2, Cabernet Sauvignon Cr3, Talia T1, Syrah S2, and, have low counts of fungi and yeasts, which are not able to develop due to the low moisture value.

The *Enterobacteriaceae* count values in the Talia T1 sample may be related to surface contamination. Considering the sample’s low moisture values, we think that this microbial development should not be considered a food safety problem.

The obtained moisture, ashes, and pH analysis (Table 2) results show that the percentage of moisture obtained is ranged between 3.37% and 9.58%, being the lowest value Arinto A4 sample and the highest 9.58% of Talia T1 sample. About the percentage of ashes, the highest value was 8.36% for the Cabernet Sauvignon Cs3 sample, and the lowest value of 4.85% for Talia T1. Finally, pH for all samples was also analyzed. cabernet C3 was the sample with the lowest pH of 3.68. In another part, Arinto A4 was the sample with the highest pH value of 4.46. It’s important to take into consideration, that all the analyzed samples presented an acidic pH.

Considering the DL No. 103/2010 [33] and NP EN ISO/IEC 17025 [34], in all the analyses performed and for the calculation purpose, the values determined below the limit of quantification (LOQ) were considered equal to half of the absolute value of LOQ.

Attending the metal content of the samples (Table 3), aluminum was the element in greatest concentration, ranging from 338.23 mg/kg (Syrah variety) and 57.33 mg/kg (Arinto variety) followed by iron that presented the highest value in Trincadeira variety (215.09 mg/kg) and the lowest value in A4 Arinto (86.59 mg/kg). As expected, mercury was the metal presented in the lowest amounts, with a mean metal/dry sample of 0.004 mg/kg in all samples included in the present work. More detailed information regarding metal concentrations present in different samples of grape pomace can be observed in Table 3.

Given the low number of samples, it is not possible to analyze whether the differences in concentrations between the different varieties are significant. However, an exploratory analysis of concentrations by type of grape (red vs. white) reveals that the differences between red grapes (*n* = 5; M2, S2, Ar2, Cs3, Tr2) and white grapes (*n* = 3; T1, A2, A4) in pH are significant (*p* = 0.036).

To understand which elements are correlated, we calculated Spearman correlations of metal concentrations. In these samples, aluminum correlates significantly with copper (Spearman’s coefficient = 0.786; *p* = 0.021) and with iron (Spearman’s coefficient = 0.929; *p* = 0.001); copper also correlates significantly with Iron (Spearman’s coefficient = 0.738; *p* = 0.037) and with zinc (Spearman’s coefficient = 0.833; *p* = 0.010).

## 4. Discussion

Regarding the results of the metal content in the different samples, it is important to consider that all the following calculations performed are focused on an adult, with body weight (bw) equal to 60 kg and independent of gender. This means that the discussion will be performed generically, not considering the different age groups from infant to elderly, nor mentioning the pregnant and breastfeeding women, who may have different nutritional needs.

Arsenic, although being classified as a nonmetal, is included in the group of heavy metals when it comes to environmental parameters. Consequently, from this point on we will roughly call arsenic a heavy metal.

Thus, considering the twelve different metal elements analyzed by atomic absorption, they can be categorized into two groups, namely essential and non-essential. The subgroup of heavy metals is included in the group of non-essential elements, while the subgroup of trace elements is included in the group of essential elements. The group of non-essential elements for human organisms containing aluminum, lithium, and nickel, and also heavy metals, which include arsenic, cadmium, lead, and mercury. The essential elements include chromium, copper, iron, manganese, and zinc, which all trace metallic elements for the human organism [35].

To achieve our objective, i.e., to understand whether the samples under study can be used safely in human nutrition, the standard parameters and values regulated by the European Commission (EC) for the different metals were considered. These EC Regulations were based on WHO’s Joint Expert Committee on Food Additives (JECFA), Food and Agriculture Organization (FAO) recommendation for the Provisional Tolerable Weekly Intake (PTWI) and Benchmark dose lower confidence limit (BMDL) recommended by European Food Safety Authority’s (EFSA’s) scientific committee for some elements that may be toxic to the human body, group where heavy metals fit [32]. In this sense, the amount of sample that can be consumed daily by an adult under the above conditions will be determined.

For heavy metals, the European Commission, through Regulation EC No 1881/2006 [36] that sets the maximum levels for certain contaminants in foodstuffs, has fixed the tolerable weekly intake (PTWI) of mercury and lead at 1.6 and 25 μg/kg bw respectively. New indications were introduced in Regulation EC No 488/2014 [37] which set the tolerable weekly intake (TWI) at 2.5 µg/kg body weight/week for cadmium. For arsenic, the estimated maximum dietary exposures BMDL01 fixed vary between 0.3 and 8 μg/kg bw/day according to EC Regulation 2015/1006 [38] annexed to the above-mentioned Regulation EC No 1881/2006 [36].

Considering the regulated values for arsenic, cadmium, mercury, and lead, the sample amounts that each individual can consume daily were determined (Table 4).

Lithium and aluminum belong to the group of non-essential elements for living organisms, but nickel, although essential for some living organisms, is also non-essential for man. Therefore, these last three metallic elements will be mentioned as non-essential.

Lithium and nickel, to date, do not have data regulated by the European Commission, unlike aluminum, which has regulations regarding PTWI, TWI, and the amount per unit of the dry mass of food product (DM).

Thus, the nickel and lithium contents do not condition the consumption of these samples, which is not the case with aluminum, whose amount of sample that can be consumed by an adult daily is expressed in Table 5.

As previously mentioned, the group of the essential metallic elements to the human organism present in this work will include Cu Cr, Fe, Mn, and Zn. These five trace elements are all mentioned in the European Commission Regulations EC 1334/2003, [37] EC 479/2006 [38], and EC 1253/2008 [39], as they are part of several compounds that are used, in animal nutrition, as food additives. As far as human nutrition is concerned, Regulations EC 953/2009 [40], and EC 1925/2006 [41] to which the addendum EC 1170/2009 [42] is added, mention substances which may be added to food, where once again Cr, Cu, Mn, Fe, and Zn are mentioned. However, it is the 2008/100/EC Directive that establishes the recommended daily allowances (RDAs). The RDAs establish the values for nutrition labeling and the calculation of what constitutes a significant amount, being 14, 10, 2, 1, and 0.04 mg for iron, zinc, manganese, copper, and chromium, respectively [43].

Considering the No Observed Adverse Effect Levels (NOAEL) used to establish upper intake levels (UL) for several elements [44] which sets the UL value of 10 mg/day for copper and 11 mg/day for manganese, the following Table 6 contains the sample amounts that can be ingested daily by adults live stage group.

Dietary zinc intake, established by WHO, IOM, and SCF, determined the UL for adults of 45, 40, and 25 mg/day [45], respectively. Based on these assumptions, the sample mass that can be daily consumed was determined and the results are present in Table 7.

Currently, there is no EC regulation regarding maximum levels of chromium, neither Cr(III), Cr(VI), nor total, in foodstuffs. However, the EFSA Panel on Contaminants in the Food Chain (CONTAM, 2014) [46] proposed a Tolerable Daily Intake (TDI) of 0.3 mg Cr(III)/kg bw. The calculations performed and shown in Table 8 take into account the assumption that all the chromium present in the samples is Cr(III).

To finalize, the additives and the food substances containing iron are regulated in the same documents previously indicated for other metallic elements. The Provisional Maximum Tolerable Daily Intake (PMTDI) for iron was defined as being 0.8 mg/kg bw [47,48,49]. Taking into account this last value, it was determined the sample amount that can be ingested daily, and the results are shown in Table 8.

Considering that all these elements can have very different origins, future studies need to try to understand which elements occur together naturally or are a result of the vine culture and winemaking process. A larger sample that also includes information on soil characteristics and products used in the winemaking process would allow us to understand this. In this way, it would be possible to inform the producers of which soils and materials should be used (or avoided) to produce wine and its by-products with the lowest concentrations of heavy metals and other non-essential elements.

## 5. Conclusions

To the best of our knowledge, the present work was the first one to specifically analyze metals, microbiological from grape pomace samples obtained from grapes that were grown in Portugal.

Taking into account the results obtained in this study, it is possible to coarsely correlate the content of microorganisms with the values of pH and humidity. However, the literature does not present microbiological limits for these products, so it was considered that all other samples have low microbial counts and are compatible with their use as raw material especially if a heat treatment is used on the final product manufacture.

Ash is the inorganic residue remaining after the samples incineration which provides a measure of the total amount of minerals within a food. Nevertheless, it was through atomic absorption that the content of twelve different metals in each of the eight samples was determined.

According to the observed results, this type of flour seems to have great potential as an ingredient for food production and/or consumption. Considering the regulations established by the European Commission, it is possible to state, according to the data obtained, that arsenic, aluminum, cadmium, and lead are the elements that condition the consumption of these samples. Except for aluminum, the remaining limiting components belong to the group of heavy metals, which can cause damage to the human body if they are not controlled. However, further studies and controls on the transport and handling of by-products are necessary, as it is not possible to assess from the data obtained whether these values come from the soils or external contamination.

## Figures and Tables

**Table 1 foods-09-01772-t001:** Microbiological analysis.

Samples	Total Bacterial Count (CFU/g)	Mold Count (CFU/g)	Yeast Count (CFU/g)	Fungi Count (CFU/g)	*Enterobacteriaceae* Count (CFU/g)
A2	1.1 × 10^2^	2.4 × 10	7	3.1 × 10	<1
A4	4.8 × 10	<1	<1	<1	<1
Ar2	1.7 × 10	<1	<1	<1	<1
Cs3	1.3 × 10	5	<1	5	<1
M2	1.8 × 10^2^	<1	<1	<1	<1
T1	9.4 × 10^4^	3	6.9 × 10	7.2 × 10	1.9 × 10^2^
Tr2	2.5 × 10^2^	<1	<1	<1	<1
S2	5	1 × 10	1.0 × 10^2^	1.1 × 10^2^	<1

**Table 2 foods-09-01772-t002:** Moisture, Ashes, and pH analyses.

Samples	Moisture/%	Ash/%	pH
A2	8.46 ± 0.10	4.35 ± 0.03	4.06 ± 0.01
A4	3.37 ± 0.41	5.33 ± 0.44	4.46 ± 0.01
Ar2	8.28 ± 0.09	5.53 ± 0.19	3.83 ± 0.01
Cs3	6.47 ± 0.05	8.36 ± 0.55	3.68 ± 0.01
M2	7.90 ± 0.09	6.98 ± 0.23	3.76 ± 0.02
T1	9.58 ± 0.09	4.85 ± 0.31	3.90 ± 0.03
Tr2	7.55 ± 0.01	6.26 ± 0.37	3.78 ± 0.02
S2	7.20 ± 0.08	7.13 ± 0.06	3.70 ± 0.01

**Table 3 foods-09-01772-t003:** Metal content (± uncertainties) in eight pomace varieties samples.

***[Metal/Sample DM]/mg/kg^−^***	**Sample** **Metal**	**Merlot** **M2**	**Talia** **T1**	**Syrah** **S2**	**Arinto** **A2**	**Aragonese** **Ar2**	**Arinto** **A4**	**Cabernet** **C3**	**Trincadeira** **Tr2**	**Mean** **(SD)**
	Al	139.10	181.98	338.82	57.17	188.24	80.29	137.04	192.21	164.36
		(±27.96)	(±36.58)	(±68.10)	(±11.49)	(±37.84)	(±16.14)	(±27.55)	(±38.63)	(86.21)
	As	0.26	0.29	0.18	0.10	0.21	0.11	0.20	0.36	0.21
		(±0.05)	(±0.05)	(±0.03)	(±0.02)	(±0.04)	(±0.02)	(±0.04)	(±0.07)	(0.09)
	Cd	0.25	0.25	0.25	0.25	0.25	0.25	0.25	0.25	
		(±0.07)	(±0.07)	(±0.07)	(±0.07)	(±0.07)	(±0.07)	(±0.07)	(±0.07)	
	Cr	2.50	2.50	2.50	2.50	2.50	2.50	2.50	2.50	
		(±0.41)	(±0.41)	(±0.41)	(±0.41)	(±0.41)	(±0.41)	(±0.41)	(±0.41)	
	Cu	30.06	87.67	48.68	12.78	22.89	6.17	16.03	34.71	32.37
		(±3.77)	(±10.99)	(±6.10)	(±1.60)	(±2.87)	(±0.77)	(±2.01)	(±4.35)	(26.08)
	Fe	119.20	177.05	202.69	106.19	187.61	86.59	160.61	215.09	156.88
		(±29.08)	(±43.20)	(±49.46)	(±25.91)	(±45.78)	(±21.13)	(±39.19)	(±52.48)	(47.48)
	Hg	0.004	0.004	0.004	0.004	0.004	0.004	0.004	0.004	
		(±0.001)	(±0.001)	(±0.001)	(±0.001)	(±0.001)	(±0.001)	(±0.001)	(±0.001)	
	Li	ND	2.50	ND	2.50	ND	2.50	2.50	2.50
			(±0.47)				(±0.47)	(±0.47)	(±0.47)	(±0.47)	
	Mn	84.12	13.00	66.89	37.02	71.32	12.56	19.12	54.50	44.81
		(±9.96)	(±1.54)	(±7.92)	(±4.38)	(±8.44)	(±1.49)	(±2.26)	(±6.45)	(28.28)
	Ni	3.75	3.75	3.75	3.75	3.75	3.75	3.75	3.75	
		(±0.60)	(±0.60)	(±0.60)	(±0.60)	(± 0.60)	(±0.60)	(±0.60)	(±0.60)	
	Pb	5.00	5.00	5.00	5.00	ND	5.00	5.00	5.00	
		(±0.81)	(±0.81)	(±0.81)	(±0.81)		(±0.81)	(±0.81)	(±0.81)	
	Zn	13.81	28.26	15.55	6.64	11.51	1.60	11.58	10.58	12.44
		(±2.01)	(±4.11)	(±2.26)	(±0.96)	(±1.67)	(±0.23)	(±1.68)	(±1.54)	(7.73)

**Table 4 foods-09-01772-t004:** Sample mass that can be consumed by day for heavy metals group.

Heavy Metal	Sample Mass/Day (g)							
	A2	A4	Ar2	Cs3	M2	T1	Tr2	S2
As	25.7–98.9	23.4–89.9	12.2–47.1	12.9–49.5	9.9–38.0	8.9–34.1	7.1–27.5	14.3–55.0
Cd	85.7	85.7	85.7	85.7	85.7	85.7	85.7	85.7
Pb	42.9	42.9	ND	42.9	42.9	42.9	42.9	42.9
Hg	3428.6	3428.6	3428.6	3428.6	3428.6	3428.6	3428.6	3428.6

ND—not determined.

**Table 5 foods-09-01772-t005:** Sample mass that can be consumed by day for aluminum.

Sample Mass/Day (g)							
A2	A4	Ar2	C3	M2	T1	Tr2	S2
149.9	106.8	45.5	62.5	61.6	47.1	44.6	25.3

**Table 6 foods-09-01772-t006:** Sample mass that can be consumed by day for trace elements.

Elements	Sample Mass/Day (g)							
	A2	A4	Ar2	C3	M2	T1	Tr2	S2
Cu	782.5	1620.7	436.9	623.8	332.7	114.1	288.1	205.4
Mn	297.1	875.8	154.2	575.3	130.8	846.2	201.8	164.5

**Table 7 foods-09-01772-t007:** Sample mass that can be consumed by day for zinc.

	Sample Mass/Day (g)							
According to	A2	A4	Ar2	Cs3	M2	T1	Tr2	S2
WHO	6777.1	28,125.0	3909.6	3886.0	3258.5	1592.4	4253.3	2893.9
IOM	6024.1	25,000.0	3475.2	3454.2	2896.5	1415.4	3780.7	2572.3
SCF	3765.1	15,625.0	2172.0	2158.9	1810.3	884.6	2362.9	1607.7

**Table 8 foods-09-01772-t008:** Sample mass that can be consumed by day for iron and chromium.

	Sample Mass/Day (g)							
	A2	A4	Ar2	Cs3	M2	T1	Tr2	S2
Fe	470.9	577.4	266.5	311.3	139.5	282.4	232.5	246.7
Cr	7200.0	7200.0	7200.0	7200.0	7200.0	7200.0	7200.0	17,200.0

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
