# Peer review of "Grape Pomace: A Potential Ingredient for the Human Diet"

_foods, 2020, doi:10.3390/foods9121772_

Round 1
Reviewer 1 Report
foods-1000100
This is a study on the safety of a grape pomace.
Major issues are:
The title is awkward. Will Spanish/Italian/American/whatever grape pomace be different? Only 4 samples were tested here, Portugal produces way more [excellent] wine than that. The Authors might want to contact a professional to suggest a much better and precise title.
The Authors state that their product is safe. This referee would be extremely careful in the absence of toxicology data.
The same line of reasoning applies to the conclusions. The Authors claim that grape pomace might be applied as an ingredient for human food. This will require a Novel Food status granted by the EFSA and the Authors do not have the ability to obtain it. Hence, this investigation should be viewed as an extremely preliminary one, without going as far as “ingredient”, “supplement”, etc. Without at least 1 million euros of investment this product will never enter the market.
Even if the product enters the market good luck with obtaining a health claim. This referee researched the literature before evaluating this paper and read with great interest PMID: 32911765. Lots of additional literature can be found in this review. In short, the evidence that grape/wine polyphenols are healthful is quite weak. Indeed, there is no EFSA health claim attributable to grape polyphenols, which says a lot.
Minor point: the Introduction is a bit too long and becomes boring to read.
Author Response
Reviewer 1
Comment 1: The title is awkward. Will Spanish/Italian/American/whatever grape pomace be different? Only 4 samples were tested here, Portugal produces way more [excellent] wine than that. The Authors might want to contact a professional to suggest a much better and precise title.
Answer: Dear sir, we appreciate your comments regarding the title of our work. This work does not focus on the application of HACCP criteria, we only performed a chemical determination of the metals as well as the contents in some microorganisms, and correlated these parameters with those established by EU Regulations. Not following the food safety criteria, we agree with the change of the title to" Portuguese grape pomace: a potential ingredient for the human diet”.
We agree with you on the quality of Portuguese wines. The composition of the wine, as well as that of the grape pomace, depends on several factors besides the genetic part, namely the geoclimatic conditions. In our work eight samples were analyzed and not four, as indicated. Of these samples, there are two that are of the same variety (Arinto). As you can see from our results (Table 3), they may not be the same just because they are of the same variety.
Comment 2: The Authors state that their product is safe. This referee would be extremely careful in the absence of toxicology data.
Answer: Dear sir, we thank you very much for your comment. As we indicated in the previous answer, the title has been changed in order not to create ambiguity.
Comment 3:The same line of reasoning applies to the conclusions. The Authors claim that grape pomace might be applied as an ingredient for human food. This will require a Novel Food status granted by the EFSA and the Authors do not have the ability to obtain it. Hence, this investigation should be viewed as an extremely preliminary one, without going as far as “ingredient”, “supplement”, etc. Without at least 1 million euros of investment this product will never enter the market.
Answer: Because we are well aware that this product is not currently considered to be a food ingredient and that this would require the Novel Food status granted by EFSA, we have begun this preliminary study and have concluded that it has the potential to be so in the future.
Comment 4: Even if the product enters the market good luck with obtaining a health claim. This referee researched the literature before evaluating this paper and read with great interest PMID: 32911765. Lots of additional literature can be found in this review. In short, the evidence that grape/wine polyphenols are healthful is quite weak. Indeed, there is no EFSA health claim attributable to grape polyphenols, which says a lot.
Answer: Dear Sir, we appreciate the comment and agree with the excellence of the article you mention. However, polyphenols are the source of several studies that we consider to be equally scientifically consistent. These types of compounds do not appear in EFSA's current health claims, but neither does this organisation indicate any contraindications. We are aware of conflicting opinions and in our work we mirror ours.
Comment 5: Minor point: the Introduction is a bit too long and becomes boring to read.
Answer: Dear sir, the introduction was written in order to contextualize the objective of our research, because we consider of great interest the recovery of grape pomace, considered as waste, so little valued today from an economic point of view.
Reviewer 2 Report
Manuscript presented for review with title: “Portuguese Grape Pomace: a Safe Potential Ingredient for the Human Diet” is really interesting and very important. The experiment was planned very carefully. I’m impress of excellent work made by authors. The Introduction section includes all necessary information about examined objects and problems.
The collected experimental material and used methods do not raise any objections.
I have one serious doubt regarding the presentation of the results. In section Materials and methods authors described statistical tools which were used for statistical elaboration of obtained results. In presented tables there is no used any statistical tools. In Tables 1 and 2 is only presented standard deviation. No p-value and letters for homogenous groups are presented. In Table 3 SD value is presented only for mean value. It should be for all values from that table. Please correct it.
Similar remark is connected with the rest of presented tables 4-8
The discussion section presents a very good comparison of the obtained results with other results available in the data basis.
The obtained conclusions are clear and in accordance with results.
Author Response
Reviewer 2
Comment 1: I have one serious doubt regarding the presentation of the results. In section Materials and methods authors described statistical tools which were used for statistical elaboration of obtained results. In presented tables there is no used any statistical tools. In Tables 1 and 2 is only presented standard deviation. No p-value and letters for homogenous groups are presented. In Table 3 SD value is presented only for mean value. It should be for all values from that table. Please correct it.
Similar remark is connected with the rest of presented tables 4-8
Answer: Dear Sir, we really appreciate your valuable comments.
The use of statistical tests was planned in the design of this project to be performed mainly for metal concentrations. Table 1 and 2 do not present data related to metal concentrations. However, due to the low number of different samples, these tests were performed with an exploratory analysis perspective. In line 184-187 we present the results of this exploratory analyses, presenting data related to the differences in pH between red and white grapes, having the caution to acknowledge that this is only an exploratory analysis. Also, Spearman correlations to find what metals tend to occur simultaneously were performed and the results are presented in lines 189-192. However, these results are not presented in any table. In the tables there is only mention to means and SD as this were the only parameters that were of interest in the elaboration of said tables. In table 3, the uncertainties values for the different metals per sample were added, having in to account that uncertainties represented were expanded uncertainties expressed at approximately the 95% confidence level using a coverage factor of k = 2.
Regarding the results contained in tables 4-8 these refer to estimated values, calculated from EU Regulations, not being experimental values
Reviewer 3 Report
This paper describes the possible application of Portuguese Grape Pomace as a safe potential ingredient for the human diet. The article is quite complete, it is of interest to the scientific community, the methods used are appropriate and the results are conveniently described. The work is interesting and deepens in the use of winemaking waste.
Line 5: “Gregorio 1,”
Line 66: Wine.
Introduction: Describe the average composition of grape pomace (amount of water, proteins, lipids, phenolic compounds, sugars ...).
Lines 101-106: Indicate which varieties are red and which are white.
Line 111: “….Spain).”
Line 117: “constant”
Line 136: “…..Inc., Cambridge, United….”
Lines 147-149: Revise units or values. They are extremely large to be mg / Kg.
Line 159: “Cs3”
Line 159: “Syrah S2,….,and”. Information is missing.
Table 2: Put a separation after and before “±”. Unify.
Lines 186-192: Put a separation after and before “=”. Put “p” and “n” in italics. Unify and apply to the entire manuscript.
Line 220: Describe “bp”
References: Put the references in the correct format of the journal. The volume must be in italics. The abbreviated words of the journal names must end with a “.”.
Author Response
Reviewer 3
Comment 1: This paper describes the possible application of Portuguese Grape Pomace as a safe potential ingredient for the human diet. The article is quite complete, it is of interest to the scientific community, the methods used are appropriate and the results are conveniently described. The work is interesting and deepens in the use of winemaking waste.
Line 5: “Gregorio 1,”
Line 66: Wine.
Answer 1: Dear Sir, we really appreciate your valuable comments. We have made the changes on lines 5 and 66 as you have indicated.
Comment 2: Introduction: Describe the average composition of grape pomace (amount of water, proteins, lipids, phenolic compounds, sugars ...).
Answer 2: Dear Sir, the nutritional characterization and determination of proteins, lipids, sugars, among other paramaters will be our next goal. We decided not to introduce in this work, where we focus essentially on the metal content. However, the determination of water amount is present in table 2 of our work, mentioned as “moisture".
Comment 3: Lines 101-106: Indicate which varieties are red and which are white.
Answer 3: The identification of the samples of red and white varaietiess were placed in section 2.7. We thank you very much for the observation.
Comment 4: Lines 111to References
Answer 4:
Line 111: “….Spain).” Barcelona, Spain are the city and the country of origin of the equipment.
Line 117: “constant”. The correction has been made, thanks for the correction.
Line 136: “…..Inc., Cambridge, United….” The correction has been made, thanks for the correction.
Lines 147-149: Revise units or values. They are extremely large to be mg / Kg. The correction has been made, thanks for the correction.
Line 159: “Cs3”. The correction has been made, thanks for the correction.
Line 159: “Syrah S2,….,and”. Information is missing. The correction has been made, thanks for the correction.
Table 2: Put a separation after and before “±”. Unify. The correction has been made, thanks for the correction.
Lines 186-192: Put a separation after and before “=”. Put “p” and “n” in italics. Unify and apply to the entire manuscript. The correction has been made, thanks for the correction.
Line 220: Describe “bp”. The correction has been made, thanks for the correction (bw).
References: Put the references in the correct format of the journal. The volume must be in italics. The abbreviated words of the journal names must end with a “.”. The correction has been made, thanks for the correction.
Round 2
Reviewer 1 Report
It’s interesting that the Authors assume I am a “sir”…
In their clumsy response they address some of the criticism yet they miss most of it. There is still no explanation re: “Portuguese” in the title, and they mention HACCP for no reason.
There is no debate on wine polyphenols and the Authors should read and cite the paper I discovered. There are other ones. The proof is: no EFSA healthclaim= no proof of activities. You cannot write anything on the label for instance. Therefore, you will not be able to use your pomace as food ingredient if you do not invest millions of euros to demonstrate safety (first) and effectiveness.
The Introduction is still verbose and boring.
Author Response
Journal: Foods
Manuscript ID: Foods-1000100
Type: Article
Title: Grape pomace: a potential ingredient for the human diet
Reviewer 1
Dear Reviewer, we thank you very much for your response and we sincerely apologize for having assumed to be a “sir”, it was not our intention at all
Comment 1
There is still no explanation : “Portuguese” in the title,
Answer 1
We have reviewed the text and fully agree with not introducing “Portuguese”, since these samples can be found in several parts of the globe where grapes are grown. We have therefore changed the title, as you have indicated.
Comment 2
There is no debate on wine polyphenols and the Authors should read and cite the paper I discovered. There are other ones. The proof is: no EFSA healthclaim= no proof of activities.
Answer 2: Dear reviewer, we read very carefully the indicated article and found very interesting the perspective and the failure correlation that exists between the effect of polyphenol compounds on human health, when studies are conducted with the respective isolated compounds. As such we have rewritted the text and introduced this article as a reference, which greatly enriches our work. We thank you once again for your contribution in improving this work.
Comment 3
Therefore, you will not be able to use your pomace as food ingredient if you do not invest millions of euros to demonstrate safety (first) and effectiveness.
Answer 3: We know that to create a novel food dossier requires a large investment, which includes the financial part. However, we think that with the publication of these results, we can give a small and humble contribution to the groups that intend to take this path.
We thank you once again for allowing us to change our paradigm regarding the specific effects of (poly)phenols on human health.

Round 3
Reviewer 1 Report
Good revision